# Chromatin Organization after High-LET Irradiation Revealed by Super-Resolution STED Microscopy

**DOI:** 10.3390/ijms25010628

**Published:** 2024-01-03

**Authors:** Benjamin Schwarz, Nicole Matejka, Sarah Rudigkeit, Matthias Sammer, Judith Reindl

**Affiliations:** Section Biomedical Radiation Physics, Institute for Applied Physics and Measurement Technology, Department for Aerospace Engineering, Universität der Bundeswehr München, 85577 Neubiberg, Germany

**Keywords:** chromatin organization, DNA repair, super-resolution microscopy, interchromatin, perichromatin

## Abstract

Ion-radiation-induced DNA double-strand breaks can lead to severe cellular damage ranging from mutations up to direct cell death. The interplay between the chromatin surrounding the damage and the proteins responsible for damage recognition and repair determines the efficiency and outcome of DNA repair. The chromatin is organized in three major functional compartments throughout the interphase: the chromatin territories, the interchromatin compartment, and the perichromatin lying in between. In this study, we perform correlation analysis using super-resolution STED images of chromatin; splicing factor SC35, as an interchromatin marker; and the DNA repair factors 53BP1, Rad51, and γH2AX in carbon-ion-irradiated human HeLa cells. Chromatin and interchromatin overlap only in protruding chromatin branches, which is the same for the correlation between chromatin and 53BP1. In contrast, between interchromatin and 53BP1, a gap of (270 ± 40) nm is visible. Rad51 shows overlap with decondensed euchromatic regions located at the borders of condensed heterochromatin with further correlation with γH2AX. We conclude that the DNA damage is repaired in decondensed DNA loops in the perichromatin, located in the periphery of the DNA-dense chromatin compartments containing the heterochromatin. Proteins like γH2AX and 53BP1 serve as supporters of the chromatin structure.

## 1. Introduction

Ionizing radiation damages living cells by the ionization of the DNA molecule itself or by the induction of reactive species that then damage the DNA molecule. Depending on the type of radiation, the amount and type of damage vary. Overall, this damage can influence the survival capacity of cells or induce carcinogenesis. The most severe of the damage is the DNA double-strand break (DSB). If the repair of this damage is defective, it leads to genetic alterations, which can then directly lead to cell death or carcinogenetic mutations. Therefore, mammalian cells have developed a multitude of response mechanisms to DSBs. The most studied ones are the signal cascades leading to repair protein accumulation and different pathways to the repair of the damage. One of the most important reactions to radiation damage is the phosphorylation of the histone variant H2AX at serine 139, forming γH2AX. This is mediated by the kinases ATM, ATR, and DNA-PK [1] in mega-base-pair (Mbp) large chromatin regions around the damage [2,3]. The γH2AX can be labeled using antibodies, which makes it visible as ionizing-radiation-induced foci (IRIF) in microscopy. Upon detection of the DSB, repair starts. It is well accepted that the repair pathway choice is crucially dependent on the DNA replication status and the complexity of damage [4]. The major pathways are homologous recombination (HR) and non-homologous end-joining (NHEJ). The first needs to rely on the sister chromatin as a template, and the second only uses end processing and ligation of ends and is therefore independent of the cell cycle. But with the discovery that each DSB is processed separately [5], more and more repair mechanisms lying somewhere between HR and NHEJ are discovered. It is assumed that the DNA end resection serves as a decisive step toward the type of repair [4]. The repair is mediated by a variety of proteins showing specific functions throughout the repair. Some are directly localized at the damage, but others cover a similar region to that of γH2AX. Also, the function can be limited to only a specific step in one repair pathway or more general during the complete repair process. Rad51, for example, is a relevant protein in homologous recombination that binds directly at the location of the resected DNA. It forms small helical scaffolds, which support the single-stranded DNA (ssDNA) ends [6]; this facilitates the homology search [7,8]. By labeling Rad51, the ssDNA can be visualized, and the direct damage location can be identified using fluorescence microscopy [9,10]. A protein binding in the Mbp large regions, where also γH2AX is formed, is 53BP1. 53BP1 is an early responder to DSB induction and is activated through ATM independent of the repair mechanism [11,12,13,14,15], where the active function of 53BP1 can only be verified through NHEJ [16]. In homology-dependent repair mechanisms, it has a structure-stabilizing role, which is supported by its inner structure [9] and its binding to chromatin remodeling factors such as EXPAND1 and PTIP [17,18]. The labeling of 53BP1 IRIF through immunofluorescence is used together with γH2AX as a dosimetry measure of low-LET radiation via the foci assay. But also in studies related to chromatin and DNA repair structure, 53BP1 plays a major role [9,10,19,20]. For example, the co-staining of 53BP1 and Rad51 showed a local exclusion at the DSB location in super-resolution STED images. Recent studies conclude that by labeling Rad51, the damage itself can be labeled; 53BP1 labels the active repair region containing the relevant proteins, and γH2AX marks the decondensed DNA surrounding the damage [9,10,20].

In recent years, it turned out that DSB repair not only depends on the cell cycle state and damage complexity but also on the location of the DSB in the cell nucleus [21,22]. Moreover, the functional and structural chromatin organization, its dynamic reorganization upon DSB induction, and its regulative influence on DNA repair define the type and efficiency of repair [23,24,25,26,27]. A well-accepted model of chromatin organization is the chromosome territory interchromatin compartment (CT-IC) model. This model is based on various microscopic studies on the distribution of different chromosomes in the cell nucleus. The chromatin in the interphase is divided into so-called chromatin domains, which correspond to the single chromosomes containing a substructure. In this substructure, three areas are defined: the chromatin territories (CTs), the interchromatin compartment (IC), and the perichromatin (PR) [28,29]. The CTs contain condensed DNA of the chromosome, which is densely packed and therefore thereby protected from toxic agents, such as reactive species. Additionally, the CT is a highly dynamic structure, as throughout the cell cycle, various regions need to be accessed by proteins, e.g., for transcription [30]. It is not yet known whether this dynamic change is due to Brownian motion or occurs through a directed movement dependent on actin, tubulin, or ATP [31,32]. In between the CTs, there are DNA-free regions connected by DNA-free channels called interchromatin compartments (ICs). The ICs are responsible for material and protein transport to varying gene loci as well as buffer storage of proteins and substrates [31]. The majority of proteins located in the IC are part of the posttranslational splicing apparatus. One of these proteins is the splicing protein SC35, which is part of the serin–arginine splicing factor (SR) family. It contains 303 amino acids and, through a highly conserved RNA-binding domain, is responsible for the processing of transcription products in the cell nucleus. It frequently occurs in the cell nucleus and is distributed in the whole IC through so-called speckles [33,34]. This location makes SC35 a perfect candidate for labeling and imaging the IC through fluorescence microscopy. The CT and the IC are separated by a <200 nm thick region called the perichromatin (PR) [9,28]. This boundary layer contains loosened DNA with active gene expression or actively repaired DNA regions. Furthermore, it is assumed that this region contains actively transcribed DNA parts in so-called transcription loops with a length of ~200 nm [35,36]. This model supports DNA mobility, which is necessary to bring transcription proteins and gene loci in the PR together. However, in this model, it is not necessary to transport a defined gene locus to a distant transcription site all across the cell nucleus. Also, from this point of view, the CT-IC model seems to be more realistic. For efficient DNA repair, the accessibility of the damaged region also plays a major role. Moreover, the detailed structure of the DNA at the time of repair is important to support proper repair and decrease the probability of mis- or un-joined DNA ends [37]. The accumulation of DNA repair proteins at the damaged DNA locus is highly dependent on the accessibility of the damage. However, the detailed structure and dynamics of DNA organization upon damage induction are not yet known and are part of ongoing research worldwide. The focus of this study lies in the identification of the chromatin structure surrounding damage induced by high-LET carbon ions to be able to clarify the connection between DNA damage location and efficiency of repair. The overlap between the IC, CT, PR, and the damage location is analyzed using immunofluorescence super-resolution microscopy. This leads to an enhanced model of chromatin organization after DNA damage induction.

## 2. Results

In this study, the complex connection between chromatin organization and DSB repair is revealed by analyzing the overlap between different chromatin structures and repair proteins using a software-based pixel-wise analysis, developed as a plugin for ImageJ (Version 1.52n), of single slices of 3D image stacks. For each combination, 50 cells from three independent experiments with three samples each were analyzed. All errors represent the standard error of the mean. Quantitative results can be found in Table 1.

The comparison of chromatin and interchromatin was performed by comparing the DNA label with the SC35 splicing factor. SC35 shows speckle-like structures in the regions with low DNA staining intensity (Figure 1a). The channels of the IC containing SC35 pervade the chromatin through the whole cell nucleus. Quantitative correlation analysis shows that (29 ± 9)% of the SC35-positive regions overlap with chromatin. This overlap is located in the border region between the two analyzed structures. By reducing the chromatin signal to the visible connections, which was performed using the skeletonize tool from ImageJ as described in the Section 4, the complex chromatin network can be visualized and the chromatin branches and their overlap with other labeled structures can be measured. The chromatin merely passes by the SC35 positive regions, and only a few ends touch the edges of the SC35 label.

In the next step, the overlap between the DNA repair region, labeled by 53BP1, and the chromatin was analyzed (cf. Figure 1b). Here again, the complex chromatin structure is visible. Furthermore, there is a visible overlap at the border of the 53BP1 foci and where single chromatin branches extend into the 53BP1-positive region. Overall, the quantitative analysis shows that (38 ± 14)% of the 53BP1-positive region overlaps with chromatin.

Next, the location of interchromatin, represented by SC35, and the region of active DSB repair, labeled by 53BP1, were compared, as shown in Figure 2a. Here, mutual exclusion is found, and only (1 ± 0.2)% of the 53BP1-positive region shows overlap with the SC35-positive region. Moreover, a buffer zone between the two signals is visible in the analysis, which could be quantified as surrounding a total amount of 134 53BP1 foci to (270 ± 40) nm thickness. This was determined by drawing an intensity profile between the two structures (53BP1 focus and SC35 signal) perpendicular to the surface of the 53BP1 focus and fitting a Gaussian function to the resulting profile (see Figure 2b). The full width at half maximum corresponds to the width of the gap.

Additionally, the connection between Rad51, which is labeling the location of the DSBs, and the chromatin was analyzed (Figure 3a). Here, an overlap of (60 ± 15)% was visible. Heterochromatin isolation, as described in the materials and methods section, enables a more detailed analysis. It was revealed that Rad51 is not located in the heterochromatic regions, as it lies in the regions with low chromatin density, visible by low intensities in the SiR-DNA signal. Furthermore, the Rad51 foci are located at the edges of these regions, which seem to border the chromatin-rich heterochromatic regions. Overall, only (10 ± 2)% of the Rad51 signal overlaps with heterochromatin, and the rest is lying in the euchromatin. Finally, Rad51 was correlated with γH2AX (Figure 3b), which labels the DNA surrounding the DSB. Again, a partial overlap with (36 ± 15)% between Rad51 and γH2AX was measured.

To complete the analysis, size measurements of the overlap between the region surrounding the damage labeled by 53BP1 and the chromatin, as well as the Rad51 IRIF, were performed in 100 overlapping regions of each IRIF. The mean size of the Rad51 IRIF was (0.12 ± 0.05) µm^2^, whereas the overlapping regions were (0.11 ± 0.07) µm^2^.

## 3. Discussion

In this short study, we quantified the overlap of different active regions in the cell nucleus after irradiation with high-LET carbon ions in a pixel-wise analysis of separate slices of 3D image stacks. The aim was to provide additional input to the model of chromatin organization after damage induction and to expand the view of chromatin remodeling.

First of all, we could show that there is a high amount of interchromatin in the cell nucleus, which is in good correspondence with studies by Rouquette et al. [38]. The analysis of SC35 as an interchromatin marker and 53BP1 as a marker for the damaged region, as well as the analysis of interchromatin and chromatin, reveal a complex network of ICs and CTs formed in the cells, exhibiting a 270 nm wide margin of loosened chromatin, which is much larger than the resolution of 105 nm. These results nicely fit a model proposed by Cremer et al. [29,39,40], where an additional region, called perichromatin, is visible in the cells. This region is meant to contain relaxed DNA accessible for transcription, replication, or repair. It is located in the periphery of the chromatin territories. In our study, Rad51 is used as a DSB marker. This is valid, as it clusters to single-stranded DNA and is responsible for homology search during repair [41,42,43]. We could show that almost no Rad51 could be found in chromatin-rich regions but rather at its border. About 90% of all overlap between Rad51 and the DNA is in regions with low DNA density and, more importantly, not in the heterochromatin, as quantified by heterochromatin reduction together with overlap analysis. Furthermore, a clear overlap between Rad51 and DNA surrounding the damage is visible, as quantified by the partial overlap with γH2AX. In correspondence with Albiez et al. [30], it can be assumed that the part of the Rad51 IRIF that does not overlap with DNA is responsible for assuring proper Rad51 supply at the damage itself.

The data, together with the existing literature about the models of chromatin organization [9,25,29,30,39,40,44], allow us to hypothesize the following model, which is also shown in Figure 4: The DNA surrounding the damage is loosened upon damage induction and recognition. This structure is supported by the phosphorylation of H2AX and the accumulation of 53BP1. For damage repair, the decondensed chromatin is stabilized as perichromatin. The DNA repair happens in the direct vicinity of the location of induction. The model of local chromatin reorganization is also supported by the fact that the visible damage track is not a straight line of IRIF, neither after ion-induced damage, as used here, nor after laser-induced damage [45,46]. An explanation of this can be that different chromatin densities influence the activation of γH2AX and the accumulation of 53BP1. Therefore, a slightly waved track occurs. The exclusion of 53BP1 and SC35 as interchromatin markers supports the hypothesis of mechanical stabilization of the damage by the accumulation of DNA recognition and repair markers [9,10,47]. The overlap of 53BP1 with single, protruding chromatin branches gives rise to the assumption that these branches contain the DSB. Also, the good correspondence of the size of the overlapping regions of 53BP1 with chromatin branches and the size of the Rad51 IRIF, which both are approximately 0.1 µm^2^, support this conclusion. The processing of the DSB by CtIP and Rif1 does not require this large amount of 53BP1 in such a large region surrounding the damage [48]. Our results, therefore, support the stabilizing function of 53BP1 during damage repair, as found in other studies [9,10,20]. A further protective role can be assumed by looking at the mutual exclusion of 53BP1 and SC35. From the literature, it is known that no splicing factors may interact with the damage [49,50]. The accumulation of 53BP1 in a large region and the consecutive formation of a large perichromatin surrounding the damage may help to ensure proper DNA repair without the interference of noxious proteins and factors. Future studies should evaluate the role of 53BP1 in this protective function, as well as the function and composition of the 270 nm buffer zone. All results were obtained by analyzing individual slices of 3D image stacks. In principle, the use of 2D slices separately rather than using the 3D structure as a whole could affect the results, but the size identified is well above the lateral and axial resolution. Therefore, we conclude that the 2D analysis does not change the results.

To sum up, it can be assumed that the DNA damage is repaired in decondensed DNA loops in the perichromatin, located in the periphery of the DNA-dense chromatin compartments. Proteins such as γH2AX and 53BP1 mechanically support the structure and protect the regions from harmful interference. This allows the processing of the damage and the sensitive single-stranded DNA in the perichromatin and helps to assure error-free DSB repair.

One challenge with this type of analysis is the limited resolution of microscopy, which limits the structures that can be analyzed by different methods. In this study, we used 3D STED microscopy, which has a lateral resolution of 105 nm, well above the measured structure sizes of interest here. Other studies using STORM imaging have even identified substructures in the radiation-induced γH2AX foci, called nanofoci, with smaller sizes down to tens of nanometers [51,52,53,54]. Although we were not able to resolve these nanofoci in this study, we believe that their existence points to another level of spatial arrangement, which is related to the way H2AX is incorporated into nucleosomes [52] and to the phosphorylation status upon damage induction [53,54], which we were not aiming to investigate here.

The growing understanding of the connection between the location of the DNA damage and the efficiency of repair mediated by chromatin structure and protein clustering helps to better understand DNA repair and radiation-induced disease patterns. In future studies, it would be interesting to add more proteins related to DNA repair into this localization picture; this would be even more interesting in other cell lines, such as fibroblasts or other tumor types. Furthermore, the temporal component is not yet well understood or investigated. For these types of studies, it would be of great interest to develop a suited live-cell model for super-resolution STED microscopy with multicolor samples.

## 4. Materials and Methods

### 4.1. Cell Culture and Irradiation

Culture of the HeLa cells (courtesy of AG Friedl, LMU Munich, Germany) was performed in an incubator at 37 °C (100% humidity, 95% air + 5% CO_2_) using RPMI medium (Sigma-Aldrich, St. Louis, Missouri, USA) containing 10% FCS and 1% penicillin/streptomycin as described previously [9]. For irradiation, 400,000 cells were seeded on (22 × 22) mm^2^ coverslips and placed in 6-well plates together with 2 mL medium on the previous day. Therefore, cells were in the exponential growth phase at the time of irradiation with 55 MeV carbon ions at the microirradiation facility SNAKE [55]. The irradiation procedure in total lasted 20 s. To keep the sample from drying, a thin layer of medium was kept on top of the cell layer. It had a thickness of d = (7.5 ± 2.5) μm, which was determined by weight measurements and calculation through the formula $d=\frac{m}{A\times \rho_{water}}$, where *m* is the mass, *A* = (22 × 22) mm^2^ is the area of the coverslip, and the density of water is $\rho_{water}=1\frac{g}{{\mathrm{cm}}^{3}}$. To obtain the defined medium thickness, the sample was taken out of the medium one minute before irradiation and dried carefully in air. The air gap between the beam exit nozzle and the sample, as well as the medium layer, caused energy loss. This led to ion energy at the cell surface of (27 ± 8) MeV and a linear energy transfer (LET) of LET = (500 ± 80) keV/μm. Irradiation was performed in a field of (3.5 × 22) mm^2^ under an angle of 9° concerning the cell layer and lasted a few seconds. The fluence of the carbon ion beam was $F=\frac{0.03}{{\mu m}^{2}}$ with a variation of 20% due to ion count rate fluctuations, which led to a dose of: $D=\frac{F\times LET}{\rho_{water}}=\left(2.4\pm 0.6\right)\mathrm{Gy}$. Per cell nucleus, there were typically two to three ion traversals visible.

### 4.2. Antibodies and Immunofluorescence Detection

After irradiation, samples were placed back in the 6-well and incubated for 1 h in the incubator. After the incubation time, cells were fixed for 15 min in a 2% (*w*/*v*) para-formaldehyde solution in PBS. This was followed by washing two times with phosphate-buffered saline (PBS), and performing three rounds of permeabilization for 5 min each with a 0.15% (*v*/*v*) Triton-X-100 solution in PBS. Then, the cells were blocked three times with PBS containing 1% bovine serum albumin (BSA) and 0.15% glycine for 10 min each, as previously described by Reindl et al. [9]. The labeling was performed with mouse anti-γH2AX (m-a-γH2AX, Sigma Aldrich, Burlington, MA, USA, #05-636, 1:350), rabbit anti-53BP1 (r-a-53BP1, Novus biologicals, BioTechne, Minneapolis, MN, USA, #NB100-305, 1:350), mouse anti-Rad51 (m-a-Rad51, GeneTex Irvine, CA, USA, #GTX70230, 1:350), or mouse anti-SC35 (m-a-SC35, Abcam, Cambridge, UK, #ab11826, 1:350) primary antibody. Samples with primary antibodies were incubated in a humid chamber at 4 °C overnight. After incubation, the samples were washed with PBS, permeabilized with Triton-X-100 for 5 min, and blocked with PBS containing BSA and glycine for 10 min. After this, the secondary antibodies were added: goat-anti-mouse or goat-anti-rabbit Alexa 488 (gam-Alexa 488 and gar-Alexa 488, ThermoFisher Scientific, Waltham, MA, USA, #A-11001 and #A-11034, 1:500) and goat-anti-rabbit Alexa 532 (gar-Alexa 532, ThermoFisher Scientific, Waltham, MA, USA #A-11009, 1:500). The samples were incubated with antibodies in a humid chamber for 2 h at room temperature. After this final incubation step, samples were washed carefully several times and mounted on glass slides using ProLong Diamond (Sigma Aldrich, Burlington, MA, USA) mounting medium. After 48 h of drying at room temperature under light exclusion, the slides were imaged and stored at 4 °C. For chromatin labeling, cells were labeled using a far-red SiR-DNA Kit (Spiochrome, Stein am Rhein, Germany). For labeling, the medium was exchanged just before irradiation and replaced with a 1000 nM solution of the Kit diluted in the medium. For the experiments, five types of co-staining were used, as shown in Table 2.

### 4.3. Microscopy

For imaging, a super-resolution optical CW STED microscope (Leica TCS SP 8 3X) was used. Excitation wavelengths 470 nm for the Abberior STAR 440SX, 514 nm for Chromeo505, and 640 nm for the SiR-DNA dye were set. The detection ranges were 473 nm to 504 nm and 518 nm to 580 nm, respectively, with a depletion laser at 592 nm for the antibodies and 650 nm to 700 nm with a STED laser at 775 nm for the SiR-DNA dye. Laser power for the excitation laser was on the order of 1 mW, and that for the STED lasers was approximately 70 mW. The STED lasers were subdivided into a lateral STED beam (40% of the energy) and an axial STED beam (60%). The temporal and temperature-dependent shift was excluded, as described previously [19]. Stacks of cell nuclei (3–5 μm thickness) with a slice distance of 160 nm were acquired with a 100x oil objective (Leica HCX PL APO 100x/1.4 Oil) and a pixel size of 40 nm. The raw data images were deconvolved using Huygens Professional (Scientific Volume Imaging, Hilversum, Netherlands) as described previously [19], resulting in a lateral resolution of 105 nm and an axial resolution of 200 nm.

### 4.4. Heterochromatin Isolation

Heterochromatin was isolated according to the value defined by Imai et al. [56]. Here, all chromatin regions were euchromatin, which has an intensity value below the threshold of $T=\frac{I_{max}}{\left(6.5\;\pm \;1.0\right)}$, where $I_{max}$ is the maximum intensity of chromatin visible in the image. Therefore, in each image, the maximum intensity was determined, and all signals were neglected below the threshold. The residual signal was considered to represent heterochromatin.

### 4.5. Overlap Analysis

For overlap analysis, the reduced product of the differences from the mean (rPDM) analysis was used for the separate slices of the 3D image stacks as described in detail by Reindl et al. [19]. This analysis is available as an ImageJ (https://imagej.net/ij/ (accessed on 27 November 2023)) software plugin for free use. Briefly, a threshold is determined above which the pixels are included for analysis. The area to be analyzed is cut out with a large margin. Then, the area of the region with a signal above a certain gray value is determined. This is performed for a range of gray values. Then the areas are compared, and a range of gray values is selected where the change in the size of the area is small in relation to the change in the gray value; i.e., a 20% change in the gray value results in a maximum 10% change in the area. After this, the rPDM value is calculated for each pixel using the following formula:$$rPDM=\left\{\begin{array}{c} \;d,\;\left|R_{i}\right.<R_{mean}ʌG_{i}<G_{mean}\\ \frac{\left(R_{i}-R_{mean}\right)\left(G_{i}-G_{mean}\right)}{\left(R_{max}-R_{mean}\right)\left(G_{max}-G_{mean}\right)},\;x\geq 0 \end{array}\right.$$
where $d\in ℜ\backslash \left[-1,1\right]$; $R_{i}$ and $G_{i}$ are the pixel values of the two channels; and $R_{mean},G_{mean\;},\;R_{max\;},\;G_{max\;}$ are the corresponding mean and maximum values. For overlap analysis, only pixels are considered that have higher pixel values than the mean in both channels. Here, two groups of pixels occur. The first groups are pixels with negative rPDM, where only one channel has a high signal; therefore, no overlap is visible. The second group with positive rPDM is the overlapping group, as there is high signal intensity in both channels. To obtain the fraction of overlapping pixels $F_{O}$, the number of pixels with positive rPDM, $P_{pos}$, is divided by the total number of pixels, $P_{tot,\;channel}$, in the relevant channel: $F_{O}=\frac{P_{pos}}{P_{tot,channel}}$. The analysis was performed on irradiated samples in all cases. In the analysis of SC35 together with chromatin staining, the irradiated region within the cells was not clearly visible. Therefore, random positions within the cell nuclei were chosen.

### 4.6. Distillation of Chromatin Network

The chromatin network was achieved by using the skeletonize tool from ImagJ software. This tool is based on the work be Lee et al. [57]. With this tool, the centerlines of the underlying network are determined in the analyzed images. For this, binary images are used, where all signals above the threshold determined for the correlation analysis are set to 255 and all other signals are set to 0. The skeletonizing is applied to these binary images, as automatically implemented in ImageJ. The remaining skeleton is considered the chromatin network.

## Figures and Tables

**Figure 1 ijms-25-00628-f001:**
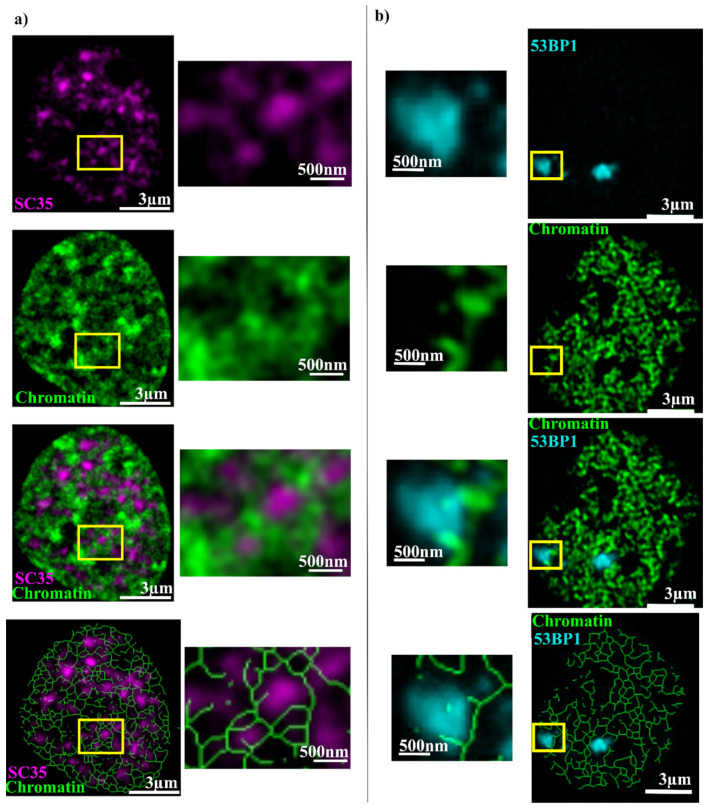
(**a**) HeLa cell with labeled chromatin (green) and SC35 (magenta) as an interchromatin marker. The first row shows SC35, the second row chromatin, the third row the overlay, and the fourth row the chromatin network together with SC35. The right column shows a zoom of the yellow box. (**b**) HeLa cell with labeled chromatin (green) and 53BP1 (cyan). The first row shows 53BP1, the second row chromatin, the third row the overlay, and the fourth row the chromatin network together with 53BP1. The chromatin network was achieved by using the skeletonize tool from ImageJ on the signal above the threshold used also for correlation analysis. The left column shows a zoom of the yellow box. No threshold is used for the image data shown.

**Figure 2 ijms-25-00628-f002:**
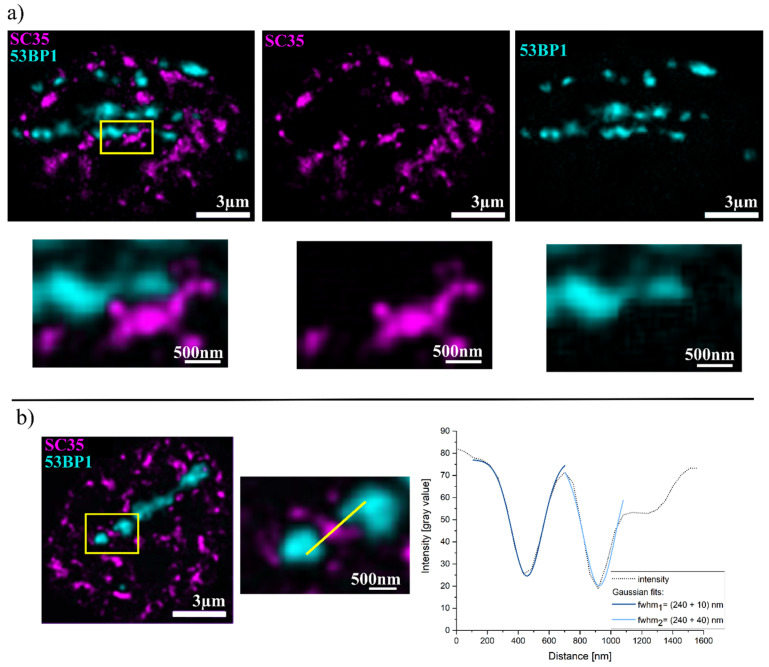
(**a**) HeLa cells with SC35 interchromatin labeling (magenta) and 53BP1 labeling of carbon-ion-induced damage (cyan). The first column shows the overlay, and the second and third columns show SC35 and 53BP1, respectively. The second line shows the zoom of the yellow box. (**b**) Example analysis of the buffer zone. IRIFs are enlarged, and a line is drawn between 53BP1 and SC35, perpendicular to the 53BP1 signal surface. The figure shows the analysis of two separate buffer zones within the ROI. The intensity profile is shown on the right. The two dips in intensity were analyzed using a Gaussian fit. No threshold was used for the image data shown.

**Figure 3 ijms-25-00628-f003:**
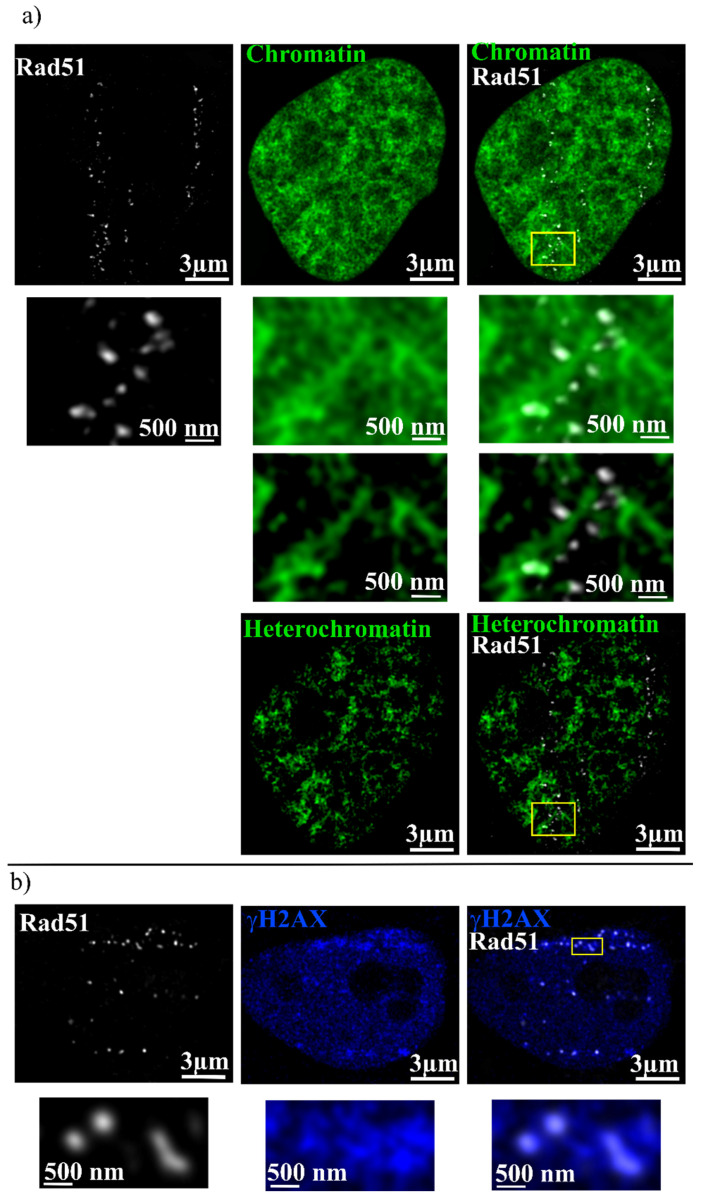
(**a**) Irradiated HeLa cell with chromatin (green) and Rad51 (gray) labeling. The first column shows Rad51, the second chromatin, and the third the overlay. The second row shows a zoom of the yellow box. In the two next rows, only the isolated heterochromatin of the same cell in combination with Rad51 is shown. (**b**) Irradiated HeLa cell showing Rad51 (gray) and γH2AX (blue). The first column shows Rad51, the second γH2AX, and the third the overlay. The lower row shows the zoom of the yellow box. No threshold is used for the image data shown.

**Figure 4 ijms-25-00628-f004:**
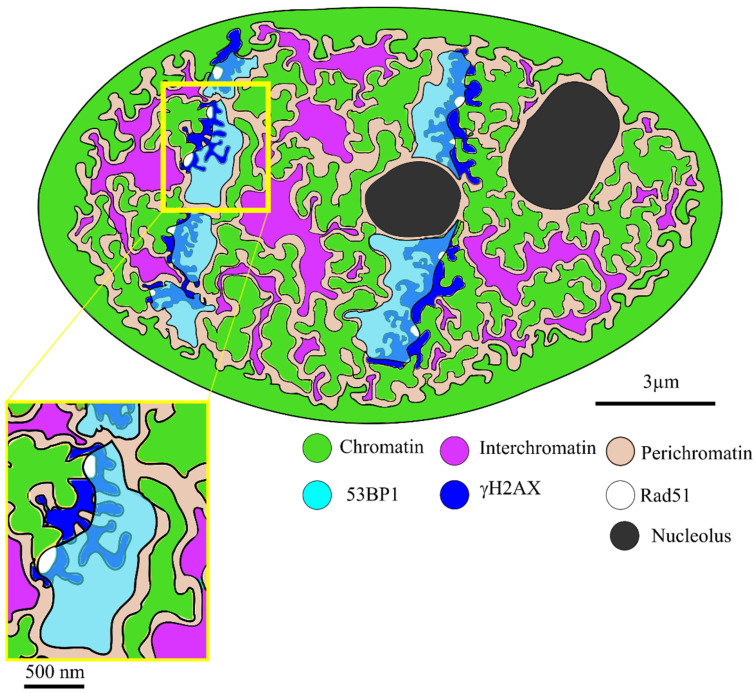
Schematic model of chromatin organization in a cell irradiated with high-LET particles. It is based on the conclusions drawn from this study and already accepted models of chromatin organization.

**Table 1 ijms-25-00628-t001:** Overlapping regions in the various combinations of staining for irradiated regions in all cases except SC35 + Chromatin, where random regions within the cell nuclei were chosen.

Combination	Overlap %
SC35 + Chromatin	29 ± 9
53BP1 + Chromatin	38 ± 14
Rad51 + Chromatin	60 ± 15
Rad51 + Heterochromatin	10 ± 2
53BP1 + SC35	1.0 ± 0.2
Rad51 + γH2AX	36 ± 15

**Table 2 ijms-25-00628-t002:** Staining combinations used in this study.

Co-Staining	Dye 1	Dye 2
m-a-SC35 + SiR-DNA	gar-Alexa 488	SiR 650
r-a-53BP1 + SiR-DNA	gar-Alexa 488	SiR 650
m-a-SC35 + r-a-53BP1	gam-Alexa 488	gar-Alexa 532
r-a-Rad51 + SiR-DNA	gar-Alexa 488	SiR 650
r-a-Rad51 + m-a-γH2AX	gam-Alexa 488	gar-Alexa 532

## Data Availability

The data presented in this study are available on request from the corresponding author. The data are not publicly available due to restriction of the university.

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
