# Peer review of "Chromatin Organization after High-LET Irradiation Revealed by Super-Resolution STED Microscopy"

_ijms, 2024, doi:10.3390/ijms25010628_

Round 1

Reviewer 1 Report

Comments and Suggestions for Authors

The manuscript entitled ”Chromatin organization after high-LET irradiation revealed by super-resolution STED microscopy” deals with the relative structural overlap of different proteins that play an important role in the CT-IC model. The authors focus on the spatial overlap of fluorescently labelled SC35, 53BP1, Rad51, gH2AX, and chromatin structures. Due to the typical feature size of the samples, the STED super-resolution imaging method was applied. I would recommend the manuscript for publication in Molecular Science after some corrections and modifications.

Questions and comments:

·       The studied labelled molecules clearly form 3D structures. However, the figures depict only 2D images and (if I understand correctly) the overlap values were also calculated in 2D. Please clarify this point. If the overlap values were indeed calculated for 2D images, do the authors expect different results for the 3D evaluation?

·       Table 1 lists the overlap values. Please mark such values before and after irradiation so the reader can follow the change introduced by the radiation.

·       Similar structures have already been imaged with dSTORM super-resolution microscopes with much higher spatial resolution (10 nm). A comparison of the different super-resolution methods applied for the study of the chromatin organization would be useful. For example, Figure 3b shows the gH2AX distribution. Due to the limited spatial resolution, the foci/nanofoci structure of the formation cannot be revealed, the accuracy of the calculated overlap value is questionable. How does the spatial resolution of the applied STED microscope (105 nm) affect the calculated overlap values?

Author Response

We thank the reviewer for this constructive report. Please find the answers in the attached document.

Reviewer 2 Report

Comments and Suggestions for Authors

The authors employed STEM microscopy experiments to investigate the correlation between chromatin structure and the organization of DNA double-strand break repair proteins. While the introduction is generally well-written, there are few results presented, and the understanding of chromatin compartments in the context of this research is quite perplexing. Consequently, I would advise against publishing this work in the International Journal of Molecular Sciences without major revision.

Major issues:

The figures lack clear definitions for the three chromatin compartments. Specifically, there is insufficient information about the intensity thresholds corresponding to chromatin territories, the interchromatin compartment, and the perichromatin region in the chromatin image (Sir650 staining).

Figure 1 lacks the definition of a chromatin network. What exactly is a chromatin network? It remains unclear whether it corresponds to the highest-intensity Sir650 staining region or chromatin territories. If it corresponds to the highest Sir650 staining region, it is crucial to define the threshold.

In Figure 2b, the method for determining the orientation and selection of two 53BP1 foci to draw for establishing the buffer zone is inadequately explained. A detailed procedure is essential for clarity.

Figure 3 introduces the term "heterochromatin." The relationship between the three-compartment system and the heterochromatin and euchromatin systems is unclear. It is uncertain whether heterochromatin corresponds to chromatin territories.

Comments on the Quality of English Language

The English proficiency is sufficiently competent to facilitate understanding.

Author Response

We thank the reviewer for this comprehensive report. Please find our detailed answers in the atatched document.
